# Child–Robot Interactions Using Educational Robots: An Ethical and Inclusive Perspective ^[note 1],[note 2],[note 3]^

**DOI:** 10.3390/s23031675

**Published:** 2023-02-03

**Authors:** Marta I. Tarrés-Puertas, Vicent Costa, Montserrat Pedreira Alvarez, Gabriel Lemkow-Tovias, Josep M. Rossell, Antonio D. Dorado

**Affiliations:** 1Department of Mining, Industrial and ICT Engineering, Universitat Politècnica de Catalunya—BarcelonaTech (UPC), 08242 Manresa, Barcelona, Spain; 2Department of Philosophy & Institut d’Història de la Ciència (iHC), Universitat Autònoma de Barcelona, 08193 Bellaterra, Barcelona, Spain; 3Artificial Intelligence Research Institute (IIIA-CSIC), 08193 Bellaterra, Barcelona, Spain; 4Research Group in Education, Neuroscience, Experimentation and Learning (GRENEA), Faculty of Social Sciences, UVic-UCC, 08241 Manresa, Barcelona, Spain; 5Department of Mathematics, Universitat Politècnica de Catalunya—BarcelonaTech (UPC), 08242 Manresa, Barcelona, Spain

**Keywords:** human–robot interaction, STEM, robotics, diversity, inclusion, robots for learning, gender, stereotypes, diversity, education

## Abstract

The Qui-Bot H_2_O project involves developing four educational sustainable robots and their associated software. Robots are equipped with HRI features such as voice recognition and color sensing, and they possess a humanoid appearance. The project highlights the social and ethical aspects of robotics applied to chemistry and industry 4.0 at an early age. Here, we report the results of an interactive study that involved 212 students aged within the range of 3–18. Our educational robots were used to measure the backgrounds, impact, and interest of students, as well as their satisfaction after interacting with them. Additionally, we provide an ethical study of the use of these robots in the classroom and a comparison of the interactions of humanoid versus non-humanoid educational robots observed in early childhood learning. Our findings demonstrate that these robots are useful in teaching technical and scientific concepts in a playful and intuitive manner, as well as in increasing the number of girls who are interested in science and engineering careers. In addition, major impact measures generated by the project within a year of its implementation were analyzed. Several public administrations in the area of gender equality endorsed and participated in the Qui-Bot H_2_O project in addition to educational and business entities.

## 1. Introduction

Project Qui-Bot provides four educational robots (Lab Qui-Bot, Lab2 Qui-Bot, Multiarm Qui-Bot, and 3D Qui-Bot) and the hardware needed to assemble them, in addition to software for controlling and programing the robots and different activities adapted to the Spanish curriculum. A technical description of the robot’s construction (sensors, motors, pieces, and 3D design) with HRI features was provided for potential users, with the software to interact with the robots by using a web page. The design of the robots and activities was realized by considering gender equality and by performing activities in the classroom in accordance with a correct ethical code. The experience was evaluated by means of specific inquiries answered by girls and boys aged between 3 and 18 years old after performing the activities. The analysis of the interaction between children and robots plays a relevant role. The robotic interaction consists of two types. First, we examined children’s interactions with robots by using computers; then, we provided feedback on children’s direct interactions with robots.

### 1.1. Goals

The main goal of this work is to develop an educational tool for contributing to the incorporation and improvement of computational thinking in pre-school, primary, and secondary school via experimentation in chemistry and robotics by using games and active actions. The work focuses on the design of methodologies and activities, testing, and validations for implementing computational thinking throughout different educational stages. The project includes the definition of a set of activities that seek to strengthen STEM (science, technology, engineering, and mathematics) competencies. The design of the Qui-Bot activities takes advantage of the innate curiosity of boys and girls toward scientific experiments from an early age, and their attraction toward playing with robots. In this manner, the content of a curriculum with respect to these subjects is complemented during different educational stages with the involvement and complicity of the teaching staff. For this reason, the proposal includes training activities that teachers and instructors can reinforce in order to guarantee success in the implementation of the project on a regular basis.

The actions of the Qui-Bot H_2_O project are aligned with the Sustainable Development Goals (SDG): quality education, gender equality, industry, innovation, and infrastructure and reduced inequalities.

### 1.2. Motivation

The great demand for professionals in IT, telecommunications, and technology and industry by companies within these sectors in the coming years, according to national studies (such as the InfoJobs report, [1], and the Barcelona Chamber of Commerce 2020, [2]), makes it necessary to develop strategies in order to increase the number of STEAM professionals. There is a persistent gender gap in STEAM university studies. Women represented only 24.8% of engineering and architecture students during the 2018–2019 academic year, and this is 4% lower than the previous year according to the latest data and figure reports from the Spanish university system (Spanish Ministry of Universities, 2020, [3]).

Several universities with specialization in technology and science (UPC, UAB) and pedagogy (UManresa, UVic-UCC) have been integrated into a collaborative transversal project, and economic and social actors (administrations, business fabric, and citizenship) have also been integrated in accordance with the guidelines in the report of the Committee of the Regions on 9 July 2019 “Strengthening STEAM education in the EU” [4] and “Bridging the gender GAP in STEM”, 2022, [5]. The consortium came together with one objective in common: the promotion of education and scientific-technical vocations from an early age, especially in the case of girls, thereby reducing the imbalance between the need for professionals and the lack of vocations.

Robotics is now widely used in educational projects, such as Lego Mindstorms or Lego competitions (see, for example, [6,7,8]). Mechanical engineering projects and computer programming are the main components of these activities. The application of robotics to other disciplines, such as chemistry, constitutes a major gap in Spanish educational policy. By combining chemistry and programming via robotics, we aim to attract boys and girls toward STEM studies in early childhood. Our study draws on the experience of the Engineering faculty at the Spanish UPC Campus Manresa and educational innovation initiatives related to chemistry.

### 1.3. Related Work

Our previous work in [9,10,11] arises from a very positive international experience based on Lego Liquid Handlers that enable Do It Yourself (DOY) construction run by the Department of Bioengineering and led by the researcher Ingmar H. Riedel-Kruse [12,13,14] at Stanford University. Alternatively, OpenLH [15] offers block-based programming interfaces for DOY liquid handlers. Opentron [16] is another open-source option. Opentron’s price is still relatively high, which discourages its use in science education. Additionally, liquid handling machines can be controlled through familiar user interfaces. The biofluids in [17], for example, were scanned and printed using a desktop printer. However, in [12,13,14,15,16,17], no gender considerations are made in the design of the software interfaces. In our work, we present open-source robots that can be easily created from scratch at low cost, enabling users without programming experience to easily automate a wide range of experiments. The entire software interface design has been analyzed by early childhood education experts to avoid reproducing gender stereotypes, while emphasizing simple visual representations and easy-to-use features. Activities are adapted with respect to the Spanish chemistry and technology academic curriculum from early childhood education to secondary education (see [18,19,20]). Validation activities have been carried out successfully. Additionally, our research includes an ethical code for the use of robots in the classroom, as a result of the tested activities and an analysis of children’s interactions with each type of robot.

Based on [12,13], we made the following improvements in the design of the liquid handling devices Lab and Lab2 Qui-Bot. Cuvette holders and extra LEGO pieces not included in the basic Mindstorms EV3 kit were designed and manufactured in 3D at a low cost to build the Lab Qui-Bot. Robots are also controlled via a simple remote control using an infrared sensor. Additional HRI features such as a loudspeaker that verbalizes the color result of the color sensor are added. Through a Bluetooth wireless connection, communication with robots is possible in real-time using a PC, tablet, or phone. Lab/Lab2 Qui-Bot software does not use Lego’s rapid interface creation tool (Lego Mindstorms EV3 Home edition software) used by [12,13]. Lab and Lab2 Qui-Bots are monitored and programmed via a web interface, tailored to attract both boys and girls to computational thinking. A software installation kit has been developed to make it easy for teachers to replicate the activities in their classrooms. The software implementation of Lab/Lab2 Qui-Bot is provided and can be updated and adapted to meet the educational center’s needs. Planned activities, unlike those conducted in [12,13], allow the learning of computational languages by pressing a sequence of buttons that must be thought of a priori to be carried out (block-based programming). These buttons correspond to instructions that the robot executes in real time and the results of the instructions are very visual. As a result of previous tests developed in [9,10,11], in our research, we developed a new Lego-based mobile robot prototype that performs chemical experiments and includes other sensing elements that allow voice recognition through simple and easy-to-use commands to facilitate robot–child interactions (MultiArm Qui-Bot). On the other hand, the connection between the robot and the computer has been improved through the use of Wi-Fi, which allows faster interactions and the monitoring of the robot, together with concurrent access. A Google Play app is available to control and program Lab and Lab2 Qui-Bots. We have also developed a humanoid prototype called 3D Qui-Bot, a non-Lego-based robot. It is possible to implement this robot at a lower cost, it can be controlled by colored blocks without requiring external devices, and it has been validated by users. Furthermore, the present research focuses on children’s behavior and human–robot ethical interactions. This study examined human–computer interactions (Lab/Lab2/MultiArm Qui-Bots) and human–robot interactions (3D Qui-Bot) at an early age. The results of this study are based on the responses of 212 children, ages 3 to 18, who had an opportunity to interact with the four Qui-Bots, as well as the responses of 64 children who had the opportunity to interact with the 3D Qui-Bot and the Lab Qui-Bot (via computer interaction). Furthermore, the robots were developed with a maker philosophy and free software, so they can be acquired by even small centers. Other instructors can analyze teaching and learning behavior as well as replicate robots using our open-source tools [21].

Additionally, this work considers ethical issues related to robotics, such as the digital divide and gender disparities. Teachers from several education centers participated in workshops and provided feedback on the implementation of Qui-Bot in the classroom. The dissemination of the materials was performed by notable female representatives in STEM fields. These include the founders of Yasyt Robotics [22], a social robotics company and a leading voice recognition device provider to assist people with disabilities. Lastly, the project’s impact on society and the economy has been discussed with references to videos of experiences, tests, mass media, and social media, e.g., tweets. 

### 1.4. Teamwork

The Qui-Bot H_2_O team comprised 15 entities and institutions following the quadruple helix model (open innovation 2.0) and 22 members (59.1% women). The preparation and testing of the activities were developed, from a technical point of view, by chemical engineering, electronics, ICT and mathematics research faculties of the Department of Mining, Industrial and ICT Engineering [23] of the Universitat Politècnica de Catalunya—BarcelonaTech (UPC) [24]. The pedagogical adaptation for the age groups was carried out with the collaboration of experts in an education, experimentation, and learning research group (GRENEA, [25]) from the Faculty of Social Sciences in Manresa (UVic-UCC) [26].

## 2. Materials and Methods

Two of the four Qui-Bot robot prototypes presented here (Lab Qui-Bot, [27], and Lab2 Qui-Bot, [28]) were inspired by the works of Ingmar H. Riedel-Kruse [12,13] from Stanford University. In contrast to the two robots developed in [12,13], our approach involves creating ad hoc software that is not Lego-based. Additionally, an app that commands the robots was developed [28,29] in addition to incorporating 3D-printed components in the construction of the robot in order to lower costs.

Using the four robots and the designed activities, students can perform chemical experiments. These experiments reproduce small-scale robots used in Industry 4.0 that simulate basic industrial processes, without introducing risks and dangerous situations. The automated experiments were designed to help students understand basic chemical concepts, such as the reflection of light, state changes, and chemical reactions such as oxidation and fermentation. These experiments complement the academic curriculum of different educational levels. 

### 2.1. Lab Qui-Bot Robot Hardware

Lab Qui-Bot is the first robot created for the project, and it can move in one dimension to store liquids in buckets. Its primary function is to interact with liquids by using an integrated syringe. The syringe can move up or down to suction liquids, and a bucket slides left or right (see Figure 1).

The Lab Qui-Bot robot can be built from a single Lego Mindstorms EV3 kit, and additional parts, such as an infrared sensor, RGB sensors, 2 large servo motors (pipette module and top module), and 1 medium servo motor (moves the robot cart)—which contains internal rotation sensors—were used. The design includes a 1 mL syringe, 20 standard 4.5 mL PMMA cuvettes, and a 3D-printed bucket holder. The sensor and infrared beacon allow us to remotely control the robot. The color sensor is used to reset the robot and detect the colors in the buckets. In addition, a microSHDC card, USB connection, and hardware using the Bluetooth Low Energy protocol were required. Figure 1 shows the main components. The RGB sensor allows discriminating the color resulting from a mixture of colors and is used to relocate the robot to the first pipette for analysis. The infrared sensor allows the robot to be controlled, which includes actions such as making it move to the right or left or raising and lowering the syringe and the syringe plunger. In contrast with the works of Ingmar H. Riedel-Kruse, [9,10], the final Lab Qui-Bot design includes the 3D design of the bucket holder and the 3D design of the additional pieces not included in the Lego Mindstorms EV3 kit. Additional information about the Lab Qui-Bot design and related software is available at [11,21,27], and information can be downloaded for use by other instructors from [21]. 

### 2.2. Lab-2 Qui-Bot Robot Hardware

Lab2 Qui-Bot (see Figure 2) evolved from Lab Qui-Bot H_2_O, with the added ability to move in two dimensions (forward, backward, right, and left). Thus, it has a much wider range of movements, enabling its interaction with a broader number of cuvettes and permitting its involvement in more complex experiments. As in the previous case, the syringe can also move up and down in order to suction liquids. Moreover, it has the ability to run different games by using several boards of different sizes. The Lab Qui-Bot robot can be built from a Lego Mindstorms EV3 kit and a Lego EV3 Education Expansion Set. The technical details of the motors, sensors, parts, and steps for construction are available at [21].

### 2.3. The Lab Qui-Bot Software and the Lab2 Qui-Bot Software

#### 2.3.1. Web Interface

The software needed to control the robot is made ad hoc and uses open-source tools. The interface of the robot is programmed with MicroPython EV3, which allows a greater number of interactions with the sensors and a development process tailored to a specific web environment that is more intuitive to use than the generic Lego Software for this application. The web interface is developed with Framework Flask [30] and CSS framework w3.css [31]. It is a cross-platform program that is compatible with operating systems such as Linux, Android, MAC, iPhone, and Windows, among others, and also with the latest versions of the most used and well-known browsers such as Firefox, Edge, and Google Chrome. Access to the web interface requires a login via the robot’s IP address (Lab Qui-Bot and Lab2 Qui-Bot) or access via a QR code. The Lab Qui-Bot and the Lab2 Qui-Bot are both monitored by a web application, which sends interaction orders to them by using Bluetooth technology as a medium for sending data and using Sockets as a communication protocol. The interact tab is a control panel that allows the user to control the robot directly by using buttons (see the details in [11,27]). Finally, the control panel on the program tab (see Figure 3) allows the user to play a set of instructions from a program and focus on the concepts of computational thinking: instruction sequence, loop, and computer logic. The built-in program is configured and displayed on the screen, and the results of the execution of the program are reproduced ipso facto, and the robot performs visual actions. 

The interface provides three pre-programmed chemical experiments that are adaptable toward educational stages: dissolution series, mixing primary colors, and density layers. More detailed information on pre-programmed experiments and technical details can be found in reference [27]. The software that allows interactions with both robots can be downloaded from [21].

#### 2.3.2. App Interface

Using the newly developed app software, the robot can be monitored remotely via Wi-Fi. As a result, a static IP address was assigned to each robot to ensure that it always has the same IP address; thus, the connection is automatic. A Wi-Fi connection was achieved with the addition of a USB antenna to the Brick EV3.

This server is powered by a Raspberry Pi 4, which can serve a Web API that controls the robot and interacts with the control application. From the application, commands can be sent to the robot using the Web API’s routes. Furthermore, it can send different data corresponding to different users, states, etc. To create an access point, the Raspberry Pi 4’s Wi-Fi interface was used. In this manner, the robot and the control device will be able to communicate with each other (using the server as an intermediary). The architecture adapts to the control of both robots (Lab and Lab2 Qui-Bot), but it does not allow their simultaneous use. In this architecture, Wi-Fi technology is the primary means of communication. By using it, all involved devices are interconnected via a single access point. Moreover, it enables widespread use, since most mobile devices are currently Wi-Fi compatible. Finally, the control device is any mobile Android-based device. Detailed technical information about the development of the command app is available at [28]. The Android application was developed using the Java programming language and is available on the Google Play Store [29]. 

Figure 4, Figure 5, Figure 6, Figure 7 and Figure 8 show the new interface used for monitoring, programming, and playing with Lab Qui-Bot and Lab2 Qui-Bot.

### 2.4. MultiArm Qui-Bot Robot

The MultiArm Qui-Bot (see Figure 9 and Figure 10) can be built by using a Lego Mindstorms EV3 kit and EV3 Education Expansion Set, a medium EV3 servo motor, an EV3 touch sensor, a color sensor, and two Lego cable packs, as well as 3D-printed parts. A Wi-Fi antenna (Edimax EW-7811 Wireless N Nano USB 150Mbps) and Raspberry Pi 4 are also required. The details of the user, assembly, and technical manuals are available at [32].

There are several 3D-designed parts that require printing: (1) two bases to hold the glasses for liquids depending on the experiment and (2) special buckets for the gripper. Paint and silicone waterproofing must be applied to these buckets. The details of how to complete these tasks are detailed in [32] in addition to the design and 3D-printing operations. The main additional materials required are as follows: (1) PLA, 1.75 mm diameter, 1 kg; (2) primary color sprays (red, yellow, and blue); (3) waterproof silicone or resin; (4) instant glue; (5) a pipette of 50 milliliters; and (6) a container that does not exceed 4 cm in height for pouring liquids.

An additional web interface is available to command and program the robot. With the pipette module, experiments involving high volume liquid mixtures using the gripper to tip the cuvettes can be conducted. Alternatively, the experiments can be conducted by using a smaller volume mixture and involves high precision by using a syringe. The gripper experiment proposes creating color mixtures from scratch from the primary colors (red, yellow, and blue). A web interface can either be used to control the robot automatically or manually. The robot searches for and deposits the colors for mixing operations in an empty container using voice commands when the user selects a color to produce from the basic colors (lilac, green, and orange). Moreover, the manual mode allows students to program the robot’s movements at a high level by using internal pre-programmed functions. The experiments with the pipette module are the same as those in Lab Qui-Bot and Lab2 Qui-Bot. A pipette automatically mixes the primary colors of two beakers. MicroPython has been used to program robot controls. React and Node.js were used to develop the server and front end, respectively. Figure 11 shows the command and programming interface of the MultiArm Qui-Bot. 

The assembly instructions were designed with Studio 2.0 and Adobe Illustrator. To design and print the 3D parts, Fusion 360 2.0.1 software from Autodesk was used along with PrusaSlicer software. A Prusa i3 MK3S+ printer was used for this project. These additional parts can be replicated on other printers by using the generated .SLT files. The programming of the experiments was performed via the Grafcet tool.

The robot with the gripper is programmed to move right, left, and center, and it can pick up a beaker, hold it, and lift it. It will tip the beaker to allow liquids to be transferred to another beaker and lower the beaker again and can be left in a chosen position. The pipette can perform actions such as suctioning the liquid with the pipette and releasing it afterwards. Two pre-programmed gripper/syringe mixing experiments have been set up either manually or automatically by voice. Technical documentation can be downloaded from [21,32].

### 2.5. Three-Dimensional Qui-Bot Robot

The 3D Qui-Bot robot is designed for children to experiment with basic sequential programming. It is possible to program the 3D Qui-Bot without a computer using colored wooden blocks. Each block contains a different color and represents a different action. The robot’s backpack can hold up to 12 blocks, which are executed once consumed. However, prior technical knowledge is not required to operate the robot. 

Students can easily and intuitively operate the robot in the classroom. Since the program is integrated into the robot, it does not require the installation of any additional programs or devices. Colored blocks were used to program the robot. Students can either add blocks one at a time or fill in the programming block by up to 12 blocks at a time. As the blocks are read sequentially, the robot will consume and execute them. When the robot reads a block, the screen (robot eyes) is changed according to the color of the block read, and the block is executed and ejected. On a table drawn with squares painted with black lines, the robot moves over them. Some containers with diluted colors (food coloring) are distributed around the table. There are some containers that are empty, and this allows the mixing of colors. Children program the robot by using colorful cubes (blocks) that should be placed in baskets. The number of baskets should equal the number of children. Ideally, small groups (no more than 12 students) are recommended. Our recommendation is to divide the children into four groups, with one on each side of the play area. In each group, there is a basket filled with a variety of colored blocks. R, L, and control batteries need to be connected to the robot (in that order) to begin the activity. In an automatic process, the robot searches for the initial positions of the arms and the syringe. As soon as the robot’s eyes light up, the activity can begin.

The robot is capable of performing the following actions according to the selected blocks. (1) The robot moves forward one square (red block) on the table until it reaches the vertex of the next square. (2) It can turn right (green block): The robot is rotated 90 degrees around its axis. (3) It can also turn left (blue block): The robot moves in the opposite direction relative to the green block. (4) Using the liquid pipette (yellow block), the robot advances forward to the container, lowers the arms, and suctions liquid. (5) The robot can carry out emptying processes (orange block): The previous steps are repeated, but the goal is to empty the liquid in this case. The surprise block (purple block) causes the robot to perform a random action and always returns to its original state. As an example, it can change the eye’s colors, advance a bit, and then move back to the starting position. Figure 12 shows the 3D Qui-Bot robot.

Three-dimensional printing, electronics, and Arduino IDE are required to make the 3D Qui-Bot replica. A 3D printer with a minimum size of 20 × 20 × 20 cm is also required.

All technical documentation about 3D Qui-Bot is available at [33] (GitHub project by LaCodornella), along with lists of materials, printing files for parts, and control board (PCB) manufacturing files. Details on the robot’s interior are accessible using FreeCAD (free 3D design software). The names of the parts in FreeCAD match their names in the BOM and 3D_part_list. Detailed information about 3D parts and purchased parts can be found in the BOM (the purchased parts, including screws) and the 3D parts list (the printed parts). Each part has specific details, such as printing time, the type and color of plastic, and the amount of plastic. The project can be printed on any FDM printer with a standard print volume. 

Below is a brief description of the main components of the 3D Qui-Bot. You can find all technical documentation at [33]. For motion, the following parts were used: (1) wheels: 2× NEMA 17; (2) arms: 2× NEMA 17; (3) syringe: NEMA 17; (4) block ejection: 5V servo motor. For sensors, the following parts were used: (1) color sensor: TCS34725; (2) laser distance sensor: VL53L0X; (3) line follower: 2× TCRT5000; (4) arm endstop: 2× Hall effect sensors; (5) syringe endstop: Hall effect sensor. The eyes are controlled by a matrix of RGB LEDs. Figure 13 shows the main components of the 3D Qui-Bot. 

ESP32 is used as a controller for the robot, and it is programmed using Arduino IDE. The technical documentation contains the necessary libraries to compile the program and the robot’s hardware. The program is available at [33] and forms a single project named Qui-Bot.ino.

KiCad was used to design the PCB. The PCB documentation section contains all the design files needed to send the PCB to the manufacturer. This includes the KiCad document, the 3D design, and the circuit diagram. The PCBs for our project were manufactured by Elecrow.

### 2.6. Activities Design

Educational materials and activities are designed using the following strategies. Students learn science and chemistry by playing games (game learning) and by following guidelines and competing in small challenge games. Collaborative project-based learning operations include the following: Students create robots that allow greater precision in chemical processes after seeing how they are directly applied in a chemical experiment, fostering creativity. They can improve their thinking and experimentation skills by modifying experiments based on the results of their trials and errors. By designing the robot, children learn how to think like professionals and develop better problem-solving skills.

All instructions for building the robot, software interface, and code to monitor the robot are available at the openCourseWare website (OCW) of the Qui-Bot H_2_O project for use by other instructors [21]. Activities are adaptable to early childhood education and primary and secondary education, and they are freely accessible via the OCW website.

## 3. Ethical Code

In this section, we introduce an ethical code associated with the activities proposed in the Qui-Bot H_2_O project as a list of general points characterizing the ethics to be respected by the involved teaching staff, that is, the principles and values that should govern the decisions and actions related to the use of these robots in the classroom. Three main general points comprise the backbone of the code: (i) fighting the digital and robotics divide, (ii) breaking stereotypes and gender biases, and (iii) tackling the dehumanization of education. 

### 3.1. The Digital Divide

The digital divide refers to the social and economic inequality generated by variances in the use of information and communication technologies [34], and it involves both institutional and individual concerns [35]. Furthermore, the integral and vital technology’s function points to the digital divide as a crucial factor for determining socio-economic inequalities [36]. Moreover, the ongoing and growing role of robotics makes it convenient to extend the conceptualization of the digital divide and include the robotics divide (see, for example, [35,37]). In this manner, although we do not explicitly mention it in this work, we do consider it. In addition, this ethical code focuses on the digital divide since children’s different applications of digital technological modes are relevant for the establishment of the digital divide [38]. Following [39], we consider a conceptualization of the digital divide comprising three categories.

#### 3.1.1. Access Divide

The access divide refers to the possibility that people have to afford devices necessary for accessing the resource. The four robots in the Quí-Bot H_2_O family are designed to be assembled and used in schools (and not in homes); thus, in principle, the digital access gap should not affect project activities. Be that as it may, to tackle this gap, the teacher in charge of directing the activity must be careful not to commission any work outside the center since this could accentuate the access divide.

#### 3.1.2. Use Divide

The use divide is related to the deficiency in digital skills that restrict the use of technology, which impedes reaping the technological progress benefits. In the present case, the teacher will have to provide training by considering the materials used by the robots, and training materials are available on the project’s web portal. In providing training, teachers and the student body can use the robots ethically, responsibly, and positively, and the activity will prevent digital inequalities generated by this divide.

#### 3.1.3. Quality of Use Gap

The quality of use gap refers to the quality of use and establishes a distinction between people who use technology at an elementary level and those that are able to appropriate and make expert and transformative use of technologies, including computer programming. Learning and teaching programming and ultimately implementing computational thinking in classrooms are firm and essential steps toward battle the digital inequalities generated by this third divide, i.e., the quality of use gap. The Qui-Bot H_2_O project tackles this digital divide as the teacher will try to ensure that all students master the programming of the robot presented in the classroom. 

### 3.2. Stereotypes and Gender Biases

Technological and scientific developments are never alienated relative to this socio-cultural context, so robotics and artificial intelligence will reproduce current biases and stereotypes. Among these, the gender bias stands out due to its urgency and relevance. Thus, the teaching team responsible for integrating the Qui-Bot H_2_O project into the educational curriculum should use the activities suggested in the project to break these gender stereotypes and to demystify some aspects of programming by considering the following points:(i)Visualize and demystify: The students should be taught some basics of computer programming before they begin programming. Furthermore, teachers should point out well-known female figures, such as Ada Lovelace, as well as some less noteworthy but also significant figures, such as Mary Allen Wilkes and Arlene Gwendolyn Lee. If the students are very young, retelling these narratives similarly to a tale would be sufficient. Moreover, in secondary school courses, teachers should explain and teach the reasons why, in the middle of the eighties, women programmers ended up being the exception and not the rule and complete this historical review by mentioning current active scientists, engineers, and popularizers with whom female students could identify with. Ultimately, one of the main project’s objectives comprises encouraging a critical discussion about the biases students have internalized regarding programming skills. Moreover, the discussion will begin by departing from common stereotypes and biases related to artificial intelligence and programming.(ii)Own awareness of skills: The proposed activities in the Qui-Bot H_2_O project include the presence of robots in the classroom, which encourage female students to carry out exercises that society, even today, does not encourage them to engage in. Considering the observations and results obtained in the experiments with young people aged from 3 to 13 [11], teachers should insist that the students become aware of their observed results and question themselves in light of having participated in programming a robot to find out what its vocations are.(iii)Reinterpret gender biases: It is critical to insist that algorithmic learning reproduces unconscious schemas that transmit prejudices, stereotypes, and discrimination, including those of gender biases. That said, the impact of this dialectic of sorts is negligible compared to the conglomerate of stereotypes and clichés that bombard the public (and, in particular, the students) without rest and from all sides (for example, media or audiovisual fiction). Via the Qui-Bot H_2_O project, students will be able to program robots and give instructions to them so that they will have taken a step further from the stereotypes about the supposed objectivity of algorithms and examine the biases inherent in programming.

### 3.3. Dehumanization of Education

Virtual robots for teaching music or languages, for instance, are already considered valuable aids in the classroom and can help personalize student learning. The same is the case with, for example, educational robots for introducing young people to programming, strengthening teamwork, consolidating concepts from various disciplines, or practicing surgical procedures. In the end, although it seems that widespread educational robotics is still some way off, the fact that teaching resources increasingly include technologies and algorithms with artificial intelligence already provides us with a glimpse of the future success of using educational robots. However, educational robot usage presents urgent and complex challenges [40]. Furthermore, the type of decisions taken concerning the challenges that arise with educational robotics and artificial intelligence will outline, at least in part, the vision that students will have of these disciplines, and this will be vital since, trivially, children will be the ones who will design future artificial intelligence and robotics.

An ethical challenge that has been raised with the development of educational robotics involves determining the teaching functions of robots. Therefore, some authors argued that among the different competencies’ students must acquire (cognitive, social, and emotional), educational robotics should focus mainly on cognitive ones while considering the risks involved [40]. In fact, and in general, the most delicate challenges appear in the scenario where an educational robot replaces the teaching staff, and they also depend on the type of skills that the robot would work on (cognitive, social, or emotional) [40]. However, Qui-Bot H_2_O robots center on fostering students’ cognitive skills. Indeed, let us remark that the ethical code presented in this work never proposed that robots are responsible for promoting and working on social and emotional aptitudes. Thus, the activities related to the project avoid some of the main concerns associated with these functions. To comply with the ethical code related to this principle, the teacher responsible for the activity should clarify the robots’ roles and functions.

## 4. Results

In what follows, we describe the resulting test activities of the Qui-Bot H_2_O project performed in two independent studies in small groups with the participation of a total number of 212 children (boys and girls) between March and July 2022. In the case of early childhood education, we provide the results of two experiences fulfilled in Lab 0_6: Discovery, Research and Documentation Center for Science Education in Early Childhood [41], located within the Faculty of Social Sciences in Manresa (Uvic-UCC). This first study involved 74 children who tested Lab Qui-Bot and 3D Qui-Bot.

For girls and boys from 7 to 17 years, we provide an evaluation of the results achieved in three guided activities: (1) one activity devoted to girls and (2) a Codeathlon contest at Manresa Technical Museum, together with (3) testing in two educational centers. All families provided their consent to participate in the Qui-Bot H_2_O activities of the project. This second study involved 82 children. This second study involved a random sample of 82 children (out of 138 tested) who built Lab Qui-Bot from scratch and tested Lab2 Qui-Bot, 3D Qui-Bot, and MultiArm Qui-Bot.

### 4.1. Children–Computer Interactions and Robotics in Early Childhood Education

The first study was carried out with boys and girls aged between 3 and 6 years old. A total of 10 children participated in the study, with 4 boys and 6 girls. The experience was developed in groups (a mixed group with one boy and one girl (4 and 5 years, respectively), two girls (3 years both), one girl alone (5 years), and one boy alone (6 years)) and guided by the educator; each experience lasted approximately 45 min for each observation (see Figure 14). Among the possible activities, the exercise of mixing liquids was chosen to observe the resulting colors, the reverse process, and processes for obtaining a color from two other colors. The results of the evaluation can be found in our previous work [11]. In our previous research in [11], we stated that all children were 100% certain that they enjoyed the programming activities and would repeat them. 

### 4.2. Children–Robotics Interactions with 3D Qui-Bot 

The second study was carried out in boys and girls between 3 and 6 years old. A total of 64 children participated in the study, with 32 boys and 32 girls. The experience was developed in groups during eight sessions and guided by the educator; the experience lasted approximately 60 min for each observation. The actions of the children were observed and registered/recorded with a video camera for a later analysis. One complete experience can be viewed from [42] (see Figure 15). To assess their satisfaction, oral questions were asked during and after the activity. 

### 4.3. Results in Early Childhood Education

In the following section, we state the conclusions of both experiences. The actions of the children were observed and registered/recorded with a video camera for a later analysis, and oral questions were also asked at the end of the activity in order to assess the gradations of satisfaction. 

#### 4.3.1. Interactions with Lab Qui-Bot

Actions between the computer and robot did not seem to cause a problem once the commands were understood (children initially introduced some problems, but once they understood the computer commands, they were able to control and program the robot.). Some children had doubts about using the mouse and touch sensor. A girl, for example, tried use her finger to select options directly on the computer screen. Boys and girls seemed to show interest in using their hands for performing actions: grabbing pipettes, touching the screen, and touching the wheels of the machine. They showed interest in the proposal, as they were not focused (in general) on how long the activity lasted. In one case, one child was already impatient and remained distracted towards the end of the experience. In the planning section of the sequence of activities of various movements (the creation of the first program), some children found the interactions more difficult. The interaction in the group with two children in the proposal was considered appropriate, and they cooperated with each other correctly; intense communication and common planning in organizing roles between them occurred. They were 100% sure that they enjoyed it and that they would repeat the activities. They also expressed that they would enjoy the activities if the robot could walk or move in some way. One experience that struck them was that the robot named the resulting color. One of the girls verbalized that the chemical experiment robot was a machine and not a robot. This is an important aspect to consider in so far as the machine does not have any humanoid robot-like features. It does not seem to be a relevant differentiation in terms of interest in the activity between boys and girls. In response to the experience, the tool was improved so that the syringe could not move beyond its maximum threshold. Furthermore, a syringe that is inside a pipette cannot be moved right or left. A larger container and fixings were proposed to make the tool more functional. A more responsible interface has been developed that displays all the icons on one screen and has a universal design for left handers. All images in the software have been simplified to avoid the stereotypical representations of scientists. The redesign also emphasizes gender equity and diversity in science, as well as a more simplified visual representation. To make the activities more interesting, we provided examples of robots performing mixing operations, and we related the mixing operations to more practical outcomes: for example, operations such as robots dyeing clothes, using kitchen dye and Chinese ink, using chemicals that stain or bleach, and adding mixture experimentation to cuisine (fermentation type) so that children feel comfortable using machines instead of using their bare hands to mix the ingredients.

#### 4.3.2. Interaction with 3D Qui-Bot

The 3D humanoid Qui-Bot is more intuitive for children to use than Lab/Lab2 Qui-Bots, and the children quickly became familiar with 3D Qui-Bot programming codes (colored cubes). The physical realization of each instruction (an instruction and a colored cube) allows children to make logical action sequences and create complex action sequences that can be revised as they are commanded by the children via the color cubes. It also allows them to “read” (interpret in advance) sequences that have already been made regardless of whether the sequences were created by other children or by themselves. Children aged under 5 years are not as adept at anticipations, so if they create a long sequence, they tend to have difficulties in following and anticipating the complex chain of actions that robots would follow to succeed in the expected end task.

The activities that 3D Qui-Bot can realize can be considered less sedentary activities than those realized with Lab Qui-Bot due to the fact that it promotes the use of the children’s own body movements to represent/anticipate the movements and requires closer contact with the children (the previous Lab Qui-Bot required more physical and emotional distance from the machine). As 3D Qui-Bot is a more humanoid robot, it also generates more empathy or induces more socialization attempts by the children when using Qui-Bot. Children sometimes talk to the 3D robot (this was not the case with the previous Lab Qui-Bot robot). The children asked the robot to speak to them. Some children expressed interest in a robot with more movement (moving backwards).

Using 3D Qui-Bot’s ability to pick up and expel liquids, to light up its face/screen based on the liquid’s color, and to move autonomously according to instructions provided, the robot generates interest and interactions with respect to the children. A 3D Qui-Bot project is more suitable for children who are over the age of five as they understand laterality and space and have the skills to anticipate, correct, and plan programming (although children younger than five have difficulties extending beyond one cube per robot’s action). Boys are more likely to monopolize/to intervene more often in game decisions than girls in the activities, but they both enjoyed the game without any significant differences. 

Although the activities the children undertake with the 3D humanoid robot were a success, several improvements need to be made to the 3D robot’s design: (1) The accuracy of color scanning depends on the illumination of the environment (for example, during outdoor lighting, the scanner tended to confuse red and orange colors or failed to interpret blue cubes); (2) the robot’s mobility and rotation need to be improved; (3) the robot needs to be made stronger, more durable, and more robust. Some parts of the robot detached and had to be “repaired” several times. The robot’s pieces have to be durable because they will be frequently used by children. Additionally, to further develop children’s anticipatory and planning skills, mechanisms and tools need to be developed. In order to anticipate which movements the robot will make, children could place the selected cubes into boxes to make the program visible or have the instructions read to them; the robot’s movements can also be marked on a small scale with individual templates, or a second carpet or a colored column can be used so that children can anticipate the robot’s movements with their bodies. Furthermore, carpet “clippings” could be used to create more interesting challenges for the children.

### 4.4. Evaluation Results of Testing the Robots in Primary and Secondary Education

Our study evaluated the child–robot interactions of Lab/Lab2 Qui-Bot, MultiArm Qui-Bot, and 3D Qui-Bot with 138 girls and boys aged between 7 and 18 years old via the following three activities: The first one was an activity for 30 girls held on International Women’s Day in Science. The results of this experience can be found in [11]. As for the second activity, it comprised a Codeathlon competition, which was held for 50 boys and girls. In the second activity, the Lab Qui-Bot was assembled and programmed, and the Lab2 Qui-Bot, the MultiArm Qui-Bot, and the 3D Qui-Bot were used and coded. The video and press experiences are available in [43,44,45,46,47]. Finally, Multi-Arm Qui-Bot and the 3D Qui-Bot have been tested in face-to-face interactions in two schools, and fifty-eight evaluation results were collected.

All activities were contextualized via a social challenge in which the children had to help the four robots solve a specific problem. More specifically, during the course of the activity, the attendees were able to perform the following actions: (1) perform a chemical experiment in the real world based on color mixing and understand the results; (2) see how the robots were able to perform the same chemical experiment; (3) build, as a team, a replica of each of Lab Qui-Bot robot by using the numbered and detailed assembly instructions provided; and (4) program the Lab Qui-Bot robot to reproduce the experiment.

The validation was carried out by means of using direct observations of the monitors and a questionnaire with 10 questions. For assessing the user friendliness of the educational robot, we adapted the Usability Scale by [48] and the Perceived Ease of Use Scale by [49] to examine the participant’s acceptance of technical systems. In accordance with the activity interest, six of the questions were validated using Likert scales (strongly disagree, little interest, enough interest, interest, and strongly agree). A random sample comprising 82 responses was collected from 138 boys and girls. The results are shown below.

#### Analysis of Results of Testing the Robots in Primary and Secondary Education

Two questionnaires were given to students at the end of the activity. We analyzed 82 responses to the questionnaires (40 girls and 42 boys). 

##### Analysis of Primary and Secondary Education Users’ Achievements in the Activities

The aim of the first questionnaire was to rate the difficulty of the activity and the degree of satisfaction. Table 1 lists the six questions related to the project’s activities. 

The feeling of accomplishment in completing the activities and the user’s learning motivation were collected via Likert scales. Students marked their level of agreement or disagreement (1: strongly disagree; 5: strongly agree) with five predefined items. In what follows, the arithmetic means of the samples are denoted by $\bar{x},$ the standard deviations of the samples are denoted by *s*, and the size of the samples is denoted by *n*. Table 2 shows, for each question, the frequency distributions with the Likert scale items.

Table 3 shows the distribution percentages of the Likert scale items. The results indicate a strong motivation for the Qui-Bot project and robotics after performing an activity. From this point onwards, the confidence interval with 95% confidence will be denoted by CI. The confidence interval according to Table 2 for question Q2 (overall rating of the course) on a scale of 1–5 is (4.39; 4.94). The results show a very high appreciation in terms of the degree of satisfaction of the activity carried out. The children rated the intention to repeat the activity (Q6) with a CI of (4.18; 5.25). This satisfaction is higher than their initial expectations. They identified previous interest (Q1) in the activity with a CI of (4.07; 4.68). Moreover, they highlighted a very high interest in repeating the chemical experimentation activities using the Qui-Bot robotics project, although they recognized that the activities were not trivial. The ease of the activity (Q4) was rated with a CI of (3.26; 3.85). Moreover, finally, the users would have liked to be able to dedicate more time to the activity (Q3), and this reflected a CI of (3.73; 4.5). Q5 indicates that most students have previously engaged in some form of robotic activity (with a CI of (3.4; 4.49)). Therefore, based on the responses received in Q2 and Q4, the Qui-Bot project can be seen as both innovative and attractive compared to other robotics projects.

In Table 4 and Table 5, we can see the hypothesis tests of the population’s arithmetic means (which we symbolized by µ) with respect to questions Q1 to Q6, considering the gender and age. This type of test was performed by using a Student’s *t*-test and examining previously whether the variances (symbolized by σ^2^) are the same or different, as these are the conditions that the Student’s *t*-test will use for later performances. It is therefore necessary to perform a two-variance test before performing the Student’s *t*-test. The information needed for this entire process is provided by the arithmetic means of the samples, $\bar{x}$; the standard deviations of the samples, *s*; and the sizes of the samples, *n*. When a significance level of 0.05 (α = 0.05) is below the *p*-value, we accepted the null hypothesis, meaning that the mean and variance are equal to one another with 95% confidence. We can observe that, for those under 12 years of age, there are differences in question Q5. There is one case (Q4) where the variances could not be considered equal ($\sigma_{1}^{2}\;\neq \;\sigma_{2}^{2})$ due to having a significance level of α = 0.05 (that is, with 95% confidence). However, in question Q4, although the variances cannot be considered equal, the arithmetic means are the same.

For girls and boys over 12 years or more, differences were observed in question Q3. In questions Q1 and Q6, the arithmetic means must be considered equal despite the fact that the variances are different, as indicated by the *p*-values. In summary, in all cases and in all six questions, there were no significant differences in the averages between the two gender/age groups, with the exception of Q3.

The following is a comparison of the five types of satisfaction levels of the four groups based on gender and age. In this case, it is necessary to perform a two-dimensional test for each of the cells listed in the tables below (see Table 6, Table 7, Table 8 and Table 9). 

After performing a test of proportion, a summary of the results is provided in Table 10 and Table 11. The *p*-values are presented in the tables. The dash notations mean that the hypothesis test did not yield a *p*-value, but since the proportions that will be compared are 0%, equal proportions must result. In question Q5 (“I have tried robotics before”), Table 10 shows that there is a difference in the number of boys and girls under 12 who answered “strongly disagree”, “little interest”, and “strongly agree”. The percentage of boys under 12 who have taken part in robotics activities is 31% (strongly agree) in contrast to 12.8% for girls, as shown in Table 6 and Table 8. In this age group, 46.1% of girls have never interacted with robotics (strongly disagree, 20.5%, or little interest, 25.6%), compared with 4.8% of boys.

As shown in Table 11, question Q1 (I was impatient to do the activities of the Qui-Bot H_2_O project) corresponding to the answers “enough interest” and “interest” must be treated differently for girls and boys. The same applies to question Q5 for the answer “interest” and question Q6 (I would repeat the activities) for the answers “interest” and “strongly agree”. As a result of the test, we must accept the alternative hypothesis, which is that boys and girls respond differently. Boys at a certain age are more likely than girls at the same age to have a previous interest in robotics. Boys are also more likely to have carried out robotics activities previously and to be interested in repeating robotics activities. To increase access to STEAM activities, encouraging girls to participate in robotics activities at an early age is critical.

##### Gathering Information from Primary and Secondary Education Students

In the second questionnaire, they were requested to answer the questions in Table 12. When asked what part of the activity they liked the most, the results exhibit the interest of girls toward (1) hardware related to robot assembly and (2) robot programming software and how the robot carried out the experiments. Moreover, (3) additional interest in chemical experimentation by hand was examined. The results are as follows: 47% (38/81) took an interest in mounting the robot, 33% (27/81) were interested in programming robotics and how the robot performed the experiments, and only 19% (16/81) preferred performing color mixing manually. Of the study’s population, 46% (38/81) of participants indicated that they signed up for this activity because a friend did, and 53% (43/81) cited vocational motivation such as “I want to be a scientist”, “I want to do experiments”, “I like to assemble things”, and “I am interested in robotics”. If we refer to the preferred school subjects, 58% chose mathematics, biology, or technology, and art and physical education were the star subjects in 42% of the cases. Considering the previous results of activity satisfaction, it can be inferred that the inclusion of Qui-Bot activities in science and technology subjects could result in an increase in satisfaction in STEM subjects.

Finally, 49.3% of the boys and girls responded that their favorite game was linked to video games, and 13.5% provided the following options: sports games, board games, and video games. The remaining 37.2% selected sports games. Therefore, 62.5% preferred games linked to computer programming.

### 4.5. Gathering Information from Teachers

Approximately 120 teachers from various educational stages were involved in the project, which is currently being tested in 14 centers and 10 different towns. The results of the evaluation of the project by a sample of teachers in one of the training sessions can be found at the following link [50]. The results show a satisfaction rate of 4.43/5. 

## 5. Qui-Bot’s Social, Economic, and International Impact

Via the online repository at Qui-Bot.upc.edu, [21], all materials created during the project was disseminated by using different means and are made available to the public. This was performed with the aim of promoting both formal and informal scientific education. Governance has been promoted by favoring shared responsibility between interest groups and all of the 15 entities participating in the project. The project was achieved thanks to the contributions of each entity, each with its own area of expertise. A network of collaboration that did not exist until today was established. The channels of communication and dissemination (websites and social networks) used by collaborators have had a major impact on society. 

The collaborations are summarized In the following: activities of the Bages County Council and the FormaBages project; activities of Manresa City Council and the University Council; catalog of the activities of the Center for Pedagogical Resources of the Department of Education; activities of the Aigües de Manresa Foundation; communication networks of the College of Industrial Engineers of Manresa (CETIM); communication networks of the Polytechnic University of Catalonia; communication networks of the University of Manresa-UVIC. The university library of the Manresa Campus supported the realization and dissemination of the activities, and the following are included: communication network of the Science and Technology Museum of Manresa; communication network of the lab_06 group Grenea uManresa; CodeLearn expert communication network; LaCodornella expert communication network; collaboration network of the Artificial Intelligence Research Institute (IIIA-CSIC); communication networks of UPC Manresa researchers; and other partner networks of the project, such as Yasyt Robots for providing expert in assistance toward dependent individuals.

Two promotional videos summarizing the experiences of the activities (testing, pilot test, and carrying out the activities) have been produced and published, with professional photographs incorporated into the free access OCW website: quibot.upc.edu.

There were 53 tweets created by researchers in the Qui-Bot project (@mtarresp, @toni_dorado, and others), with a total of 45.365 Twitter Analytics impressions. The purpose of the tweets was to expand the educational use of Qui-Bots in order to encourage the replication and application of HRI tools in robotics. This will enable scientific learning by using robots that spark interest in girls and boys. This was also performed to facilitate learning technical and programming concepts. 

As a summary of the publicity in the media, the Qui-Bot H_2_O Project has been publicized on television with three TV reports ([42,43,44]). Moreover, 11 articles in the national press (Regió7, NacióDigital, and ManresaDiari) were published with an estimated impact of 122.000. Press articles are available at [45]. The Qui-Bot project has been disseminated and communicated via international conferences [9,10] and publications [9,10,11]. Additionally, dissemination and communication have been carried out via 10 workshops, conferences, and face-to-face conferences (383 face-to-face attendees). A total of 20 communication materials (brochures, posters, etc.) and 6 UPC Bachelor’s theses have been produced. Workshops and communication materials are available at [21,46,47].

## 6. Discussion

The start up of the experience of Qui-Bot and the analysis of the results performed in early childhood education clearly demonstrates that boys and girls exhibited the same level of interest in the robotics activities, and children–robot interactions did not differ by gender. Both girls and boys expressed their surprise and satisfaction when the robot verbalized the color resulting from the mixture. They also expressed that they would like the robot to have more humanoid robot-like features, such being able to move or walk. 

In this sense, the 3D humanoid robot is designed to be intuitively controlled and to awaken children’s intuition when it comes to experimenting and programming. It is observed that it is easy for children to understand how the robot works. In general, after playing with the 3D Qui-Bot for a while, children tended to remember the color-based logical order of the cubes. There were times when it was necessary to use the “table” of instructions to refer to what actions each color represented. The children attempted tests with longer/more complex sequences when they understood how some cubes worked or were programmed.

The children’s interaction with the robot at times permits the verbalization of some social interaction situations: “hello robot!” and “go ahead!” The robot possesses a humanoid appearance when the child observes it: “it looks like a person”. There are questions or observations made about the robot’s physiognomy, such as “what’s in his hand?”, “the liquid goes to his stomach”, “he has some buttons on his head!”, “he’s sleeping”, or “why doesn’t he drink water?” At first, it is unclear to them what the tubes are for (they speculate if they are clamps or threads, if they are used for sucking air, etc.). A child picked up the wooden block used to program the robot, holding it up in front of its “face” as if the robot can actually see with its eyes open. The children interacted verbally when giving advice, asking questions, or formulating hypotheses. They clearly identified a robot as being cool (not only a simple machine). Frequently, the children asked complex questions, such as “What is the robot’s way of detecting color?” By programming each cube to represent a square (segment) of a carpet, the children could better calculate the number of cubes they needed so that the robot can move on. When the robot’s screen lit up with the color of the selected cube, the children anticipated the robot’s movements. The children generally collected cubes that the robot spontaneously expelled.

In the presence and assistance of an adult educator’s during the process, they observed the children’s actions related to EE.FF. (Executive Functions such as anticipating, self-correcting, working memory, and others). The adult educator attempted to get children to plan movements and to act with intention before placing the cubes in the robots. However, children are typically spontaneous and quickly placed cubes on the robot’s back without stopping to think carefully about how it would then act. During the activity, to encourage children in using EE.FF. (Executive Functions), the educator asked the children about what actions they would like the robot to perform. Thus, children can express their hypotheses, act in accordance with them (place the cubes they wanted on the robot’s back), and test if the robot’s actions confirm or contradict their previous hypotheses. In other words, by asking how the robot will act, the children made their own predictions, such as “And if we put the blue cube?” Children cooperated with each other and gave each other advice for programming the robot correctly (i.e., use of the Executive Function of self-correction by all group members).

In terms of gender interactions, boys tended to monopolize games more with the robot. The girls also participated, but they stood in the background more in contrast to how the boys would approach the robot and stood within its proximity. 

New research studies are underway to add enhanced HRI functionalities to the Qui-Bots in order to allow children and Qui-Bots to engage in novel interactions. The next steps include the management of a remote laboratory so that, for example, the robot located in the school lab can be commanded from home. We are working (see [51]) on the implementation of a web interface that can help monitor the results of the surveys and have a more significant educational impact. From a theoretical standpoint, we plan to develop a new taxonomy of educational objectives and apply Kolb’s experimental learning theory on our future work on Qui-Bot H_2_O.

## 7. Conclusions 

Filling the gap between high demands for technology professionals and the lack of technical vocations for girls is essential. With Qui-Bot, children and girls in particular can learn computational thinking, technology, and science using activities based on their innate interests in chemical experiments. The introduction of Industry 4.0 using Qui-Bot activities into the Spanish curriculum at a young age allows preventing social factors from negatively influencing children’s careers in science and engineering. The results of one-year implementation of the project show that 98.7% of elementary school students graded the activity with the highest score (66.6% scored 5/5 strongly agree and 32.1% scored 4/5 agree). Moreover, 91% of participants would repeat this activity or another one similar to it (73.7% gave a score of 5, strongly agree, and 18.4% gave a score of 4). 

As a result of previous research conducted by Ingmar H. Riedel-Kruse et al. [14], in our approach, the Liquid Handler models have been further developed, expanded and analyzed for applicability. Qui-Bot’s project is not only applicable to middle schools but also to all educational phases, including early childhood. Additionally, Qui-Bot’s model emphasizes engaging girls and boys in STEAM through the design of original prototypes with HRI features. A wide range of educators showed a high amount of interest in the implementation of Qui-Bot activities into classroom curriculums, particularly the concept of playfully teaching robotics and training skills in science and computational engineering. 

Some limitations identified by teachers in this proposal are related to the need to purchase or rent Lego Mindstorms kits in case multiple groups of students require robots at the same time. Additionally, teachers who have previously used Lego software for robotics tasks are less likely to use custom interfaces. Regardless, teachers unfamiliar with software development will appreciate the ease of installing the software and recognizing the robot via the web interface and app. Furthermore, they appreciate the intuitive nature of the interfaces. The applicability and impact of the project may be enhanced with at least two hours of interactive training. Teacher-led training may include assembling and testing the robots from scratch as part of the training process. 

The research of our approach in the early years students was remarkable, since no previous studies had been conducted in the literature on the combination of robotics and chemical experiments at this age. There is more interest in the Qui-Bot 3D humanoid model programmable without additional external computing devices than in the other models, in this case. Additional studies are expected to be conducted regarding the implantation of the 3D humanoid robot in higher education as part of a joint hardware and software project. As a result of this feedback, prototypes based on external control devices or intelligent devices that are already integrated into the robot will be able to progress further in development. Through the integration of HRI components into robots, as well as the collaboration of educational agents, we aim to increase students’ interest in science and technology, especially among girls. It is expected that the findings highlighted in this study can assist lecturers in implementing novel activities using Qui-Bot models and adapting them to the specific characteristics of schools and topics. 

There are no previous studies published by other authors that allow an exhaustive comparison of the models that we present, but our work may be replicated and make it possible for other authors to implement the designed Qui-Bots, to apply the activities considering the ethical code developed, and to compare with our results to conduct improvements and motivate future research. Other instructors and users can access all robot replicas, exercises, data, and supplementary materials of the project online [21].

## Figures and Tables

**Figure 1 sensors-23-01675-f001:**
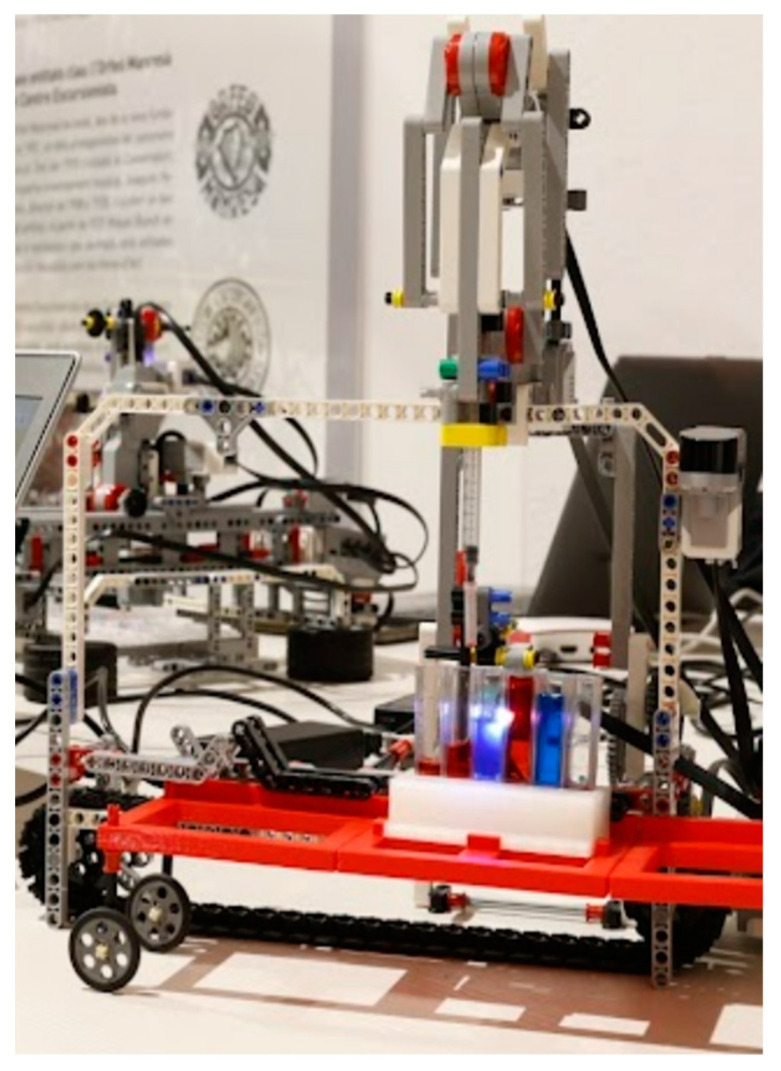
Lab Qui-Bot [11,21,27] with the RGB sensor activated (Front view).

**Figure 2 sensors-23-01675-f002:**
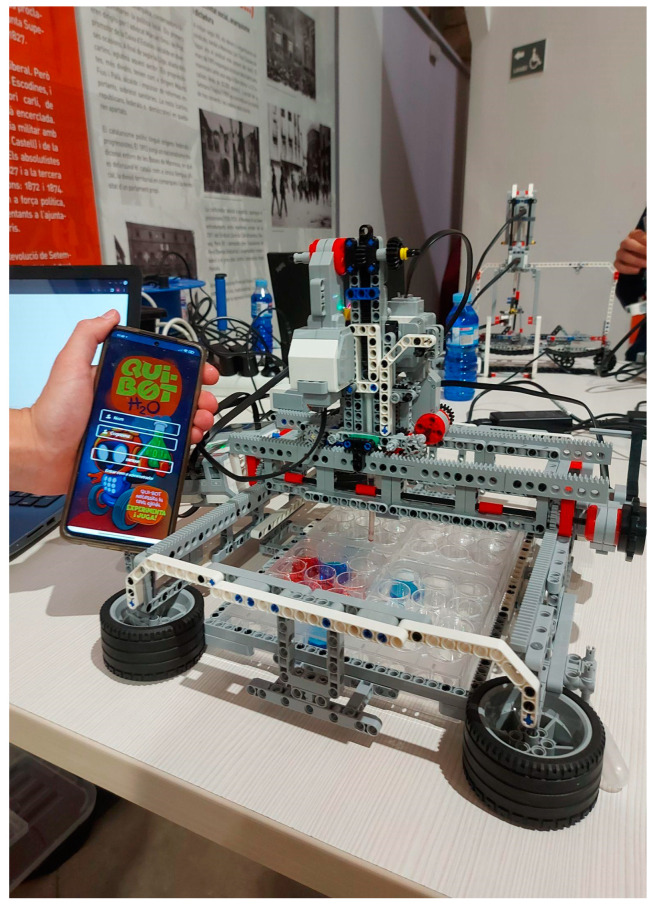
Lab-2D Qui-Bot app commanded [21].

**Figure 3 sensors-23-01675-f003:**
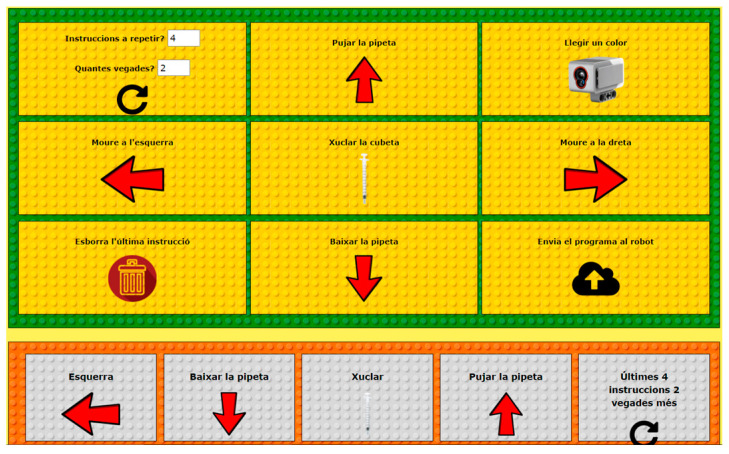
Program tab in the Control Panel where the robots are programmed [27].

**Figure 4 sensors-23-01675-f004:**
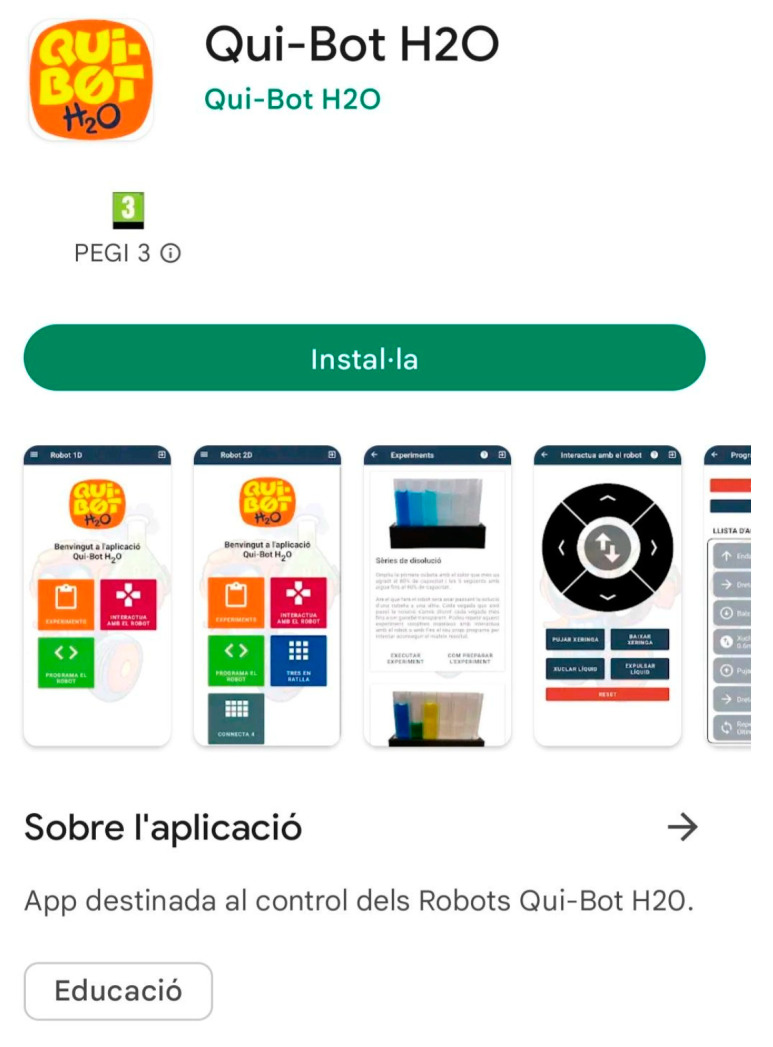
The Qui-Bot app [28,29] (Google Play).

**Figure 5 sensors-23-01675-f005:**
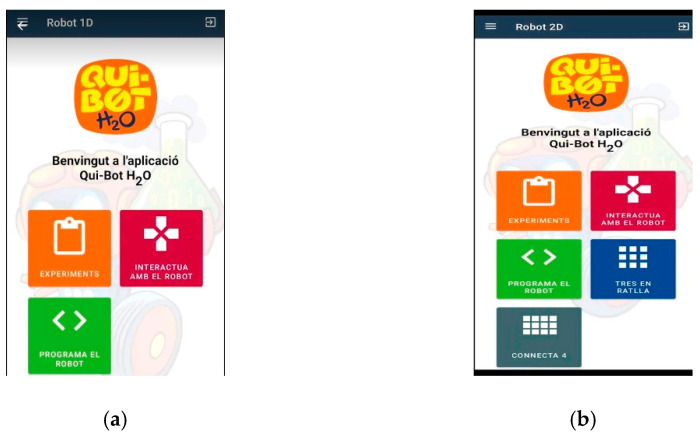
(**a**) Lab Qui-Bot (experiments, interacting with the robot, and programing the robot) and (**b**) Lab2 Qui-Bot (experiments, interactions, robot programming, tic-tac-toe game with the robot, and connect 4 game).

**Figure 6 sensors-23-01675-f006:**
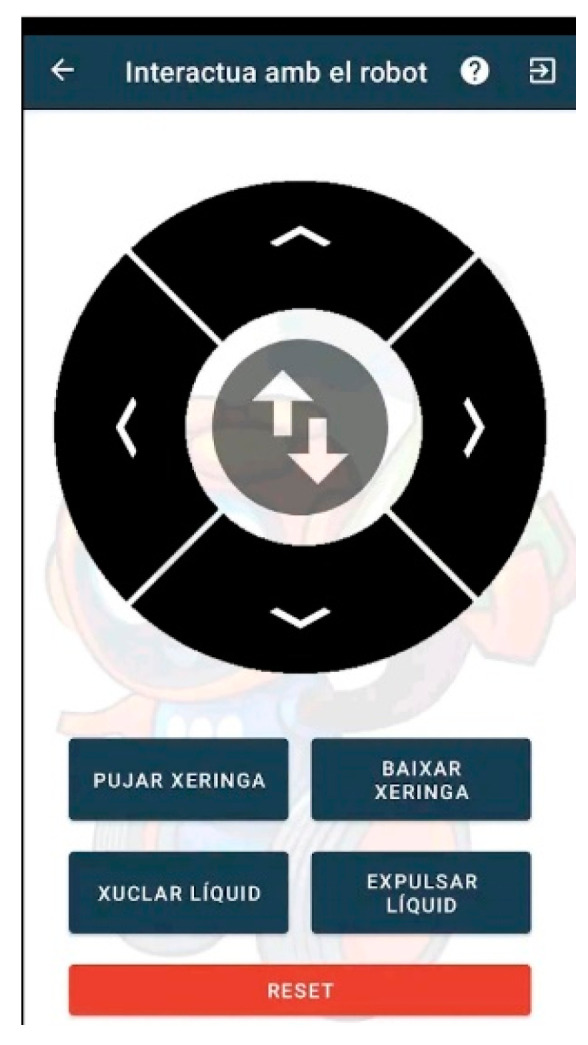
Interface for interacting with Lab Qui-Bot and Lab2 Qui-Bot.

**Figure 7 sensors-23-01675-f007:**
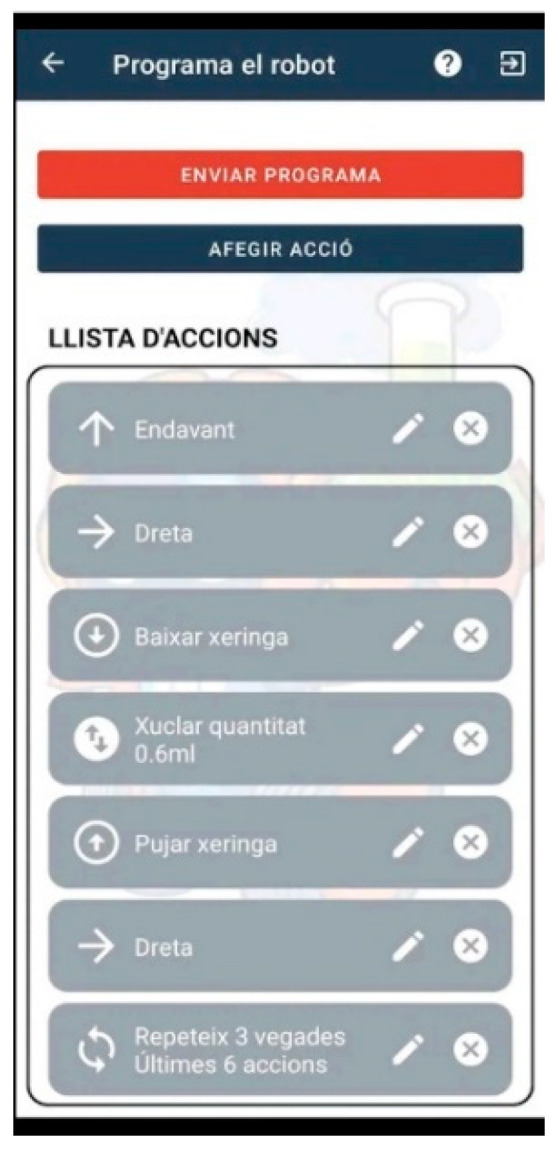
Interface for programming the Lab Qui-Bot and Lab2 Qui-Bot.

**Figure 8 sensors-23-01675-f008:**
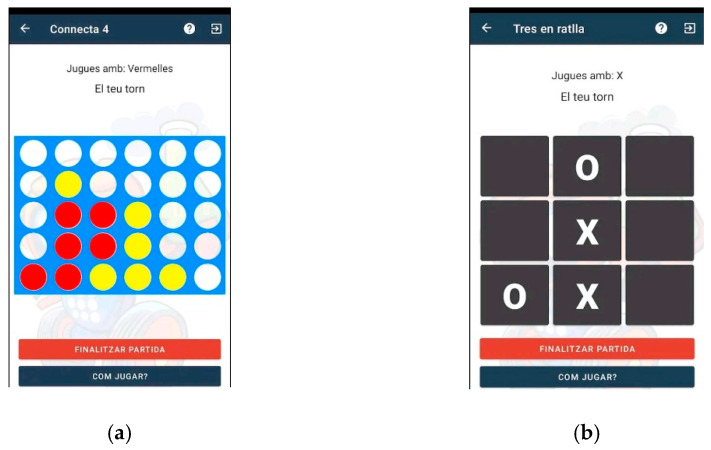
Interfaces to play with Lab2 Qui-Bot: (**a**) connect four and (**b**) tic-tac-toe.

**Figure 9 sensors-23-01675-f009:**
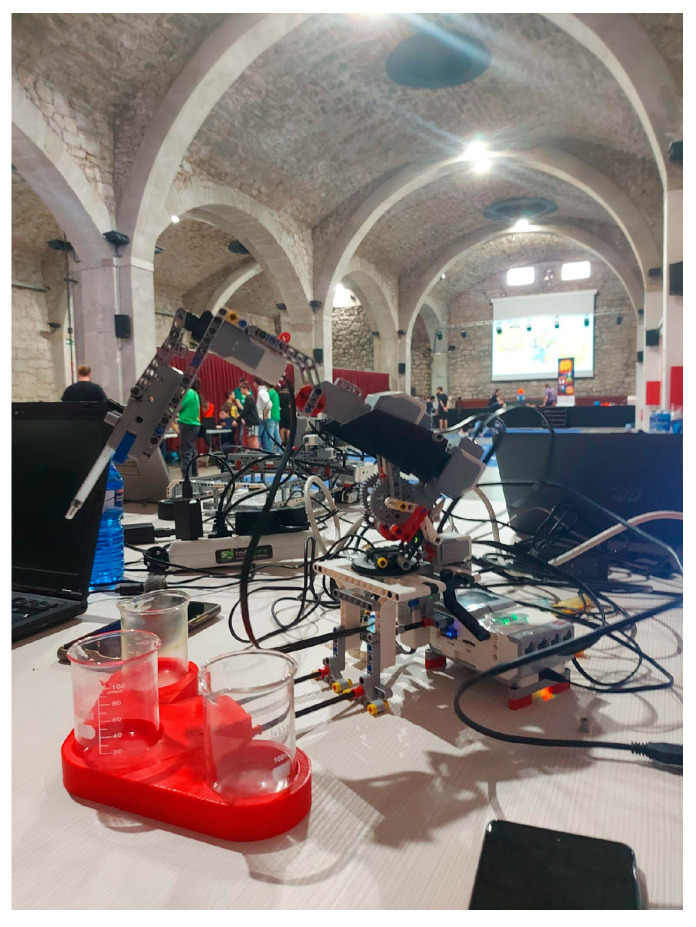
MultiArm Qui-Bot [32] with the syringe plunger for suctioning colored liquids in the 3D-designed pipette holder.

**Figure 10 sensors-23-01675-f010:**
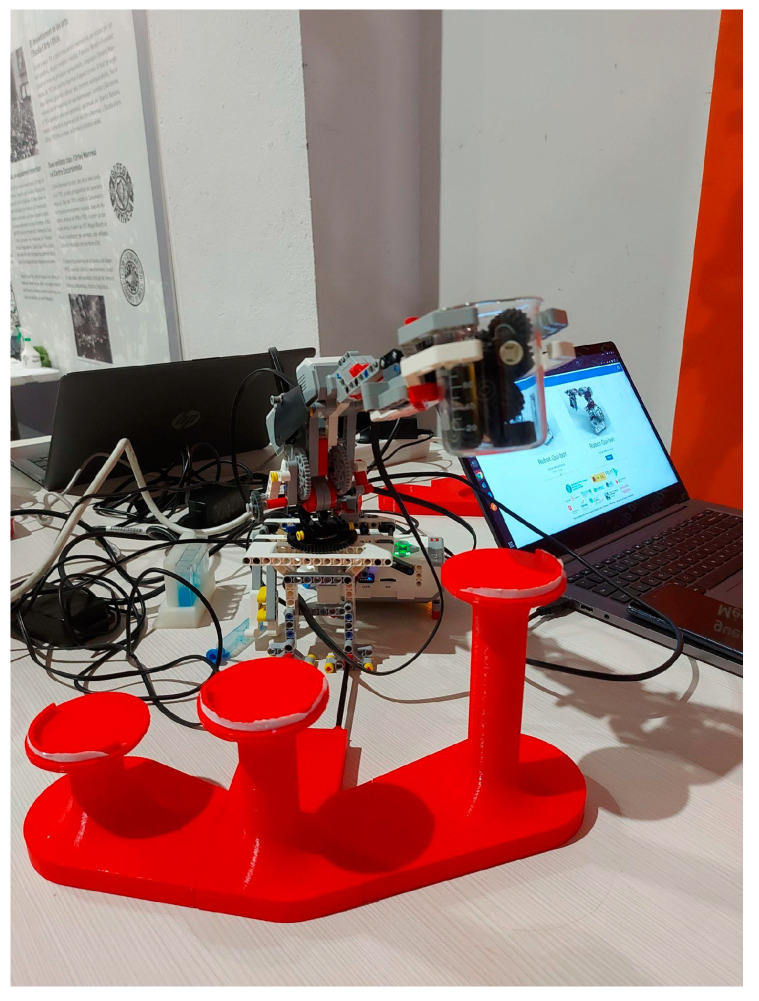
MultiArm Qui-Bot [32] equipped with a gripper for tipping cuvettes.

**Figure 11 sensors-23-01675-f011:**
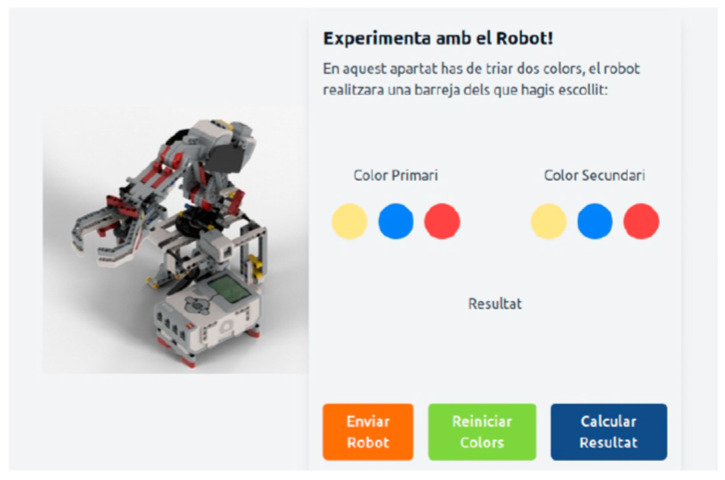
MultiArm Qui-Bot interface for controlling and programming the robot [32].

**Figure 12 sensors-23-01675-f012:**
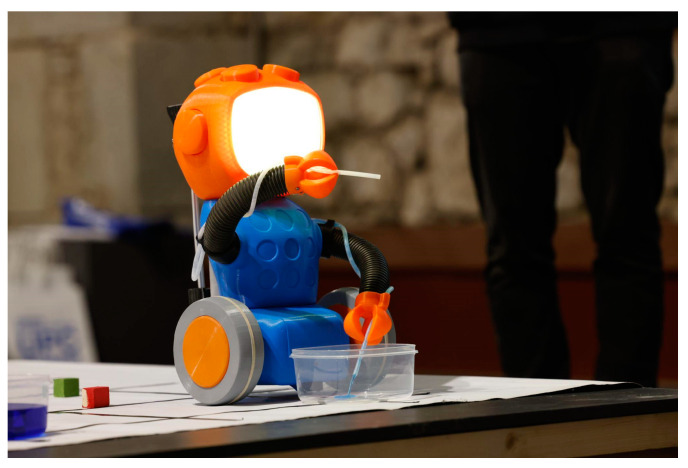
The 3D Qui-Bot, the programming blocks (green and blue) and the robot emptying blue liquid [33].

**Figure 13 sensors-23-01675-f013:**
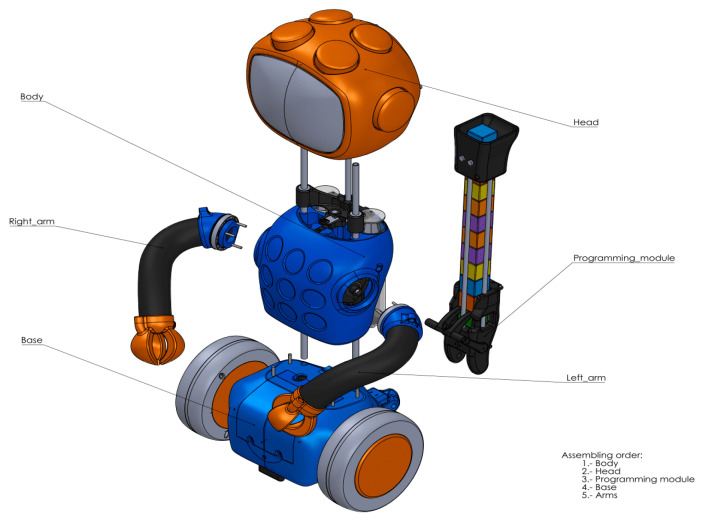
Main components of 3D Qui-Bot, LaCodornella (Assembling order: Body, Head, Programming module, Base, Left arm, Right arm) [33].

**Figure 14 sensors-23-01675-f014:**
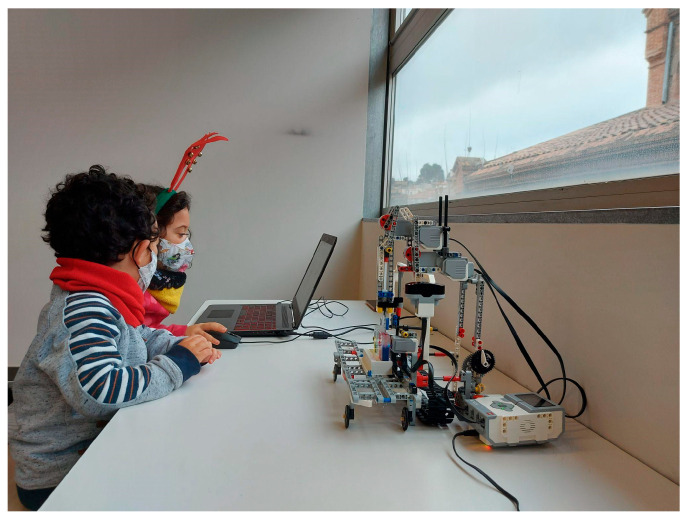
Testing Lab Qui-Bot in early ages.

**Figure 15 sensors-23-01675-f015:**
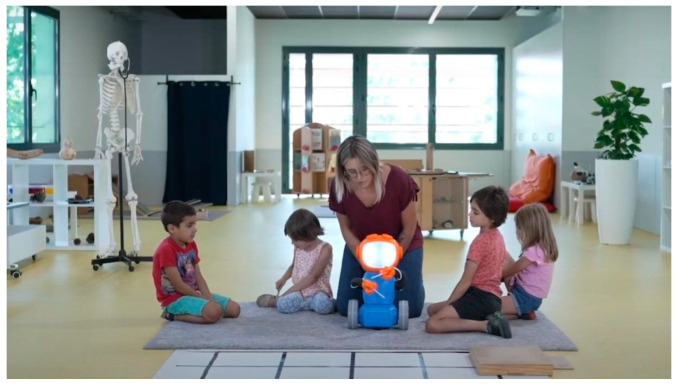
Testing 3D Qui-Bot in early ages (source Fibracat [42]).

**Table 1 sensors-23-01675-t001:** A questionnaire specifically designed to measure satisfaction with the activities.

Q1: I was impatient to do the activities of the Qui-Bot H_2_O project
Q2: I enjoyed the activities
Q3: The time to do the activities is enough
Q4: I find the activities easy
Q5: I have tried robotics before
Q6: I would like to repeat the activities of the Qui-Bot H_2_O project

**Table 2 sensors-23-01675-t002:** The frequency distribution of items on the Likert scale according to Table 1.

		Question					
Items		Q1	Q2	Q3	Q4	Q5	Q6
1	strongly disagree	0	0	2	1	10	0
2	little interest	0	0	4	4	14	0
3	enough interest	6	1	16	39	10	6
4	interest	35	26	23	31	6	14
5	strongly agree	40	54	37	7	41	56
*n*		81	81	82	82	81	76
$\bar{x}$		4.38	4.67	4.12	3.56	3.95	4.72
*s*		0.79	0.72	1.01	0.77	1.43	1.40

**Table 3 sensors-23-01675-t003:** A percentage representation of the frequency distribution of items on the Likert scale according to Table 2.

		Question					
Items		Q1	Q2	Q3	Q4	Q5	Q6
1	strongly disagree	0.00%	0.00%	2.44%	1.22%	12.35%	0.00%
2	little interest	0.00%	0.00%	4.88%	4.88%	17.28%	0.00%
3	enough interest	7.41%	1.23%	19.51%	47.56%	12.35%	7.89%
4	interest	43.21%	32.10%	28.05%	37.80%	7.41%	18.42%
5	strongly agree	49.38%	66.67%	45.12%	8.54%	50.62%	73.68%

**Table 4 sensors-23-01675-t004:** Hypothesis test of variances and population means using chi-square and Student’s *t*-test distributions for girls and boys under 12 years of age (primary education).

AGE/SEX	Q1	Q2	Q3	Q4	Q5	Q6
<12 girls	$\bar{x}$ = 4.53 s = 0.63 n = 30	$\bar{x}$ = 4.67 s = 0.55 n = 30	$\bar{x}$ = 4.29 s = 0.97 n = 31	$\bar{x}$ = 3.32 s = 0.65 n = 31	$\bar{x}$ = 2.50 s = 1.38 n = 30	$\bar{x}$ = 4.58 s = 0.64 n = 26
<12 boys	$\bar{x}$ = 4.67 s = 0.48 n = 18	$\bar{x}$ = 4.83 s = 0.38 n = 18	$\bar{x}$ = 4.28 s = 1.02 n = 18	$\bar{x}$ = 3.39 s = 1.04 n = 18	$\bar{x}$ = 4.33 s = 1.24 n = 18	$\bar{x}$ = 4.67 s = 0.56 n = 17
	*p* = 0.26 $\sigma_{1}^{2}$ = $\;\sigma_{2}^{2}$	*p* = 0.13 $\sigma_{1}^{2}$ = $\;\sigma_{2}^{2}$	*p* = 0.80 $\sigma_{1}^{2}$ = $\;\sigma_{2}^{2}$	*p* = 0.03 $\sigma_{1}^{2}$ $\neq \sigma_{2}^{2}$	*p* = 0.64 $\sigma_{1}^{2}$ = $\;\sigma_{2}^{2}$	*p* = 0.58 $\sigma_{1}^{2}$ = $\;\sigma_{2}^{2}$
	*p* = 0.45 $\mu_{1}=\mu_{2}$	*p* = 0.26 $\mu_{1}=\mu_{2}$	*p* = 0.97 $\mu_{1}=\mu_{2}$	*p* = 0.81 $\mu_{1}=\mu_{2}$	*p* = 0.00 $\mu_{1}\neq \mu_{2}$	*p* = 0.32 $\mu_{1}=\mu_{2}$

**Table 5 sensors-23-01675-t005:** Hypothesis test of variances and population means using chi-square and Student’s *t*-test distributions for girls and boys aged 12 years or older (secondary education).

AGE/SEX	Q1	Q2	Q3	Q4	Q5	Q6
≥ 12 girls	$\bar{x}$ = 4.11 s = 0.33 n = 9	$\bar{x}$ = 4.67 s = 0.50 n = 9	$\bar{x}$ = 4.33 s = 0.71 n = 9	$\bar{x}$ = 4.00 s = 0.71 n = 9	$\bar{x}$ = 4.78 s = 0.67 n = 9	$\bar{x}$ = 5.00 s = 0.01 n = 9
≥ 12 boys	$\bar{x}$ = 4.21 s = 0.72 n = 24	$\bar{x}$ = 4.50 s = 0.51 n = 24	$\bar{x}$ = 3.58 s = 1.10 n = 24	$\bar{x}$ = 3.54 s = 0.66 n = 24	$\bar{x}$ = 4.21 s = 1.25 n = 24	$\bar{x}$ = 4.54 s = 0.72 n = 24
	*p* = 0.03 $\sigma_{1}^{2}$ $\neq \sigma_{2}^{2}$	*p* = 0.98 $\sigma_{1}^{2}$ = $\;\sigma_{2}^{2}$	*p* = 0.20 $\sigma_{1}^{2}$ = $\;\sigma_{2}^{2}$	*p* = 0.73 $\sigma_{1}^{2}$ = $\;\sigma_{2}^{2}$	*p* = 0.07 $\sigma_{1}^{2}$ = $\;\sigma_{2}^{2}$	*p* = 0.00 $\sigma_{1}^{2}$ $\neq \sigma_{2}^{2}$
	*p* = 0.60 $\mu_{1}=\mu_{2}$	*p* = 0.41 $\mu_{1}=\mu_{2}$	*p* = 0.03 $\mu_{1}\neq \mu_{2}$	*p* = 0.16 $\mu_{1}=\mu_{2}$	*p* = 0.10 $\mu_{1}=\mu_{2}$	*p* = 0.07 $\mu_{1}=\mu_{2}$

**Table 6 sensors-23-01675-t006:** The satisfaction level response proportion of girls < 12 years old (primary education).

		Question					
Items		Q1	Q2	Q3	Q4	Q5	Q6
1	strongly disagree	0.00%	0.00%	0.00%	0.00%	20.50%	0.00%
2	little interest	0.00%	0.00%	5.00%	5.00%	25.60%	0.00%
3	enough interest	5.10%	2.60%	12.50%	45.00%	15.40%	5.70%
4	interest	25.60%	20.50%	15.00%	25.00%	2.60%	20.00%
5	strongly agree	46.20%	53.80%	45.00%	2.50%	12.80%	48.60%

**Table 7 sensors-23-01675-t007:** The satisfaction level response proportion of girls >= 12 years old (secondary education).

		Question					
Items		Q1	Q2	Q3	Q4	Q5	Q6
1	strongly disagree	0.00%	0.00%	0.00%	0.00%	0.00%	0.00%
2	little interest	0.00%	0.00%	0.00%	0.00%	0.00%	0.00%
3	enough interest	0.00%	0.00%	2.50%	5.00%	2.60%	0.00%
4	interest	20.50%	7.70%	10.00%	12.50%	0.00%	0.00%
5	strongly agree	12.60%	15.40%	10.00%	5.00%	20.50%	25.70%

**Table 8 sensors-23-01675-t008:** The satisfaction level response proportion of boys < 12 years old (primary education).

		Question					
Items		Q1	Q2	Q3	Q4	Q5	Q6
1	strongly disagree	0.00%	0.00%	2.40%	2.40%	2.40%	0.00%
2	little interest	0.00%	0.00%	0.00%	2.40%	2.40%	0.00%
3	enough interest	0.10%	0.00%	2.40%	21.40%	4.80%	2.40%
4	interest	14.30%	7.10%	16.70%	9.50%	2.40%	4.90%
5	strongly agree	28.60%	35.70%	21.40%	7.10%	31.00%	34.10%

**Table 9 sensors-23-01675-t009:** The satisfaction level response proportion of boys >= 12 years old (secondary education).

		Question					
Items		Q1	Q2	Q3	Q4	Q5	Q6
1	strongly disagree	0.00%	0.00%	2.40%	0.00%	2.40%	0.00%
2	little interest	0.00%	0.00%	4.80%	2.40%	7.10%	0.00%
3	enough interest	9.50%	0.00%	21.40%	23.80%	2.40%	7.30%
4	interest	26.20%	28.60%	14.30%	28.60%	9.50%	12.20%
5	strongly agree	21.40%	28.60%	14.30%	2.40%	35.70%	39.00%

**Table 10 sensors-23-01675-t010:** Results of the test of proportions between girls and boys aged <12 years old.

		Question					
Items		Q1	Q2	Q3	Q4	Q5	Q6
1	strongly disagree	---	---	0.30	0.30	0.03	---
2	little interest	---	---	0.14	0.90	0.01	---
3	enough interest	0.14	0.31	0.22	0.58	0.39	0.82
4	interest	1.00	0.40	0.15	0.44	0.73	0.20
5	strongly agree	0.64	0.22	0.58	0.15	0.00	0.20

**Table 11 sensors-23-01675-t011:** Results of test of proportions between girls aged ≥ 12 and boys aged ≥ 12.

		Question					
Items		Q1	Q2	Q3	Q4	Q5	Q6
1	strongly disagree	---	---	0.31	---	0.31	---
2	little interest	---	---	0.14	0.31	0.06	---
3	enough interest	0.03	---	0.07	0.26	0.54	0.06
4	interest	0.00	0.37	0.30	0.78	0.03	0.02
5	strongly agree	0.07	0.37	0.30	0.54	0.07	0.01

**Table 12 sensors-23-01675-t012:** Science, technology, engineering, and math (STEM)-related generic questions.

Q1: Which is your favorite game? (sport games, board games, video games)
Q2: What subject is your favorite in school?
Q3: Why have you signed up for this activity?
Q4: What activity did you like the most today?

## Data Availability

The answers to the questionnaires are available at https://t.ly/dUwbn (accessed on 24 November 2022).

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
