# Peer review of "Child–Robot Interactions Using Educational Robots: An Ethical and Inclusive Perspectiveâ€"

_sensors, 2023, doi:10.3390/s23031675_

Round 1
Reviewer 1 Report
The work is interesting and can be useful, but there is no novelty in the presented work. The authors have recently published similar work, " Tarrés-Puertas, M.I., Merino, J., Vives-Pons, J., Rossell, J.M., Pedreira Álvarez, M., Lemkow-Tovias, G. and Dorado, A.D., 2022. Sparking the Interest of Girls in Computer Science via Chemical Experimentation and Robotics: The Qui-Bot H2O Case Study. Sensors, 22(10), p.3719.". Hence, the present work is not suitable for publication.
Author Response
All the issues have been addressed and we have summarized of the brand-new
research presented in this article. Please see the attachment. Thank you very much for your help.

Reviewer 2 Report
This paper is about the construction and use of educational robots in the school context. A remarkable feature is the maker philosophy and free software behind the development of the robots, so that even centers with a small budget may access to them. Another distinguishing feature of this work is the consideration of ethical aspects, in particular the digital divide and the gender gap as for robot use. Last, but not least, the use of the robots is encouraged via the performance of experiments in chemistry.
The text is in general well-written and clearly structured, except for the sections and section numbering commented below. While there is no new break-through contribution as for sensors or perception algorithms, the paper provides a valuable insight on how to build low-budget robots and get the most of them in the educational context. I particularly value Section 3 (Ethical code), as it addresses quite relevant and pungent questions in the educational use of robots. I also appreciate the authors providing particular examples (e.g. in the children’s responses) and not just the statistical results.
There is some confusion about the test groups in Section 3. The introductory paragraph suggests that there are two main groups of experiments carried out: one for early childhood teams (74), and another one for children from 7 to 17 years of age (a sample of 82 out of 138 individuals). Section 3.1 and 3.2 describe the experiments, on 6 and 64 children respectively, which is a total of 70, not 74. The second set of experiments, described from line 605 onwards, concerns two activities, one for 30 girls in the International Women’s day in Science and the Codeathlon for 50 boys and girls, which is a total of 80 individuals, not 138. There is also an intersection in age ranges between the two groups, as in the first one there is a row of experiments carried out on children aged 6-9 (line 520). This is not important, but should be remarked. By the way, numbering of the sections is also wrong: you have two sections numbered as 3.2, in lines 518 and 605. I think the wrong ones are the first: there should be a 3.1 Section (early childhood), with a 3.1.1. (Lab Qui-bot) and 3.1.2 (3D Qui-bot and Lab Qui-bot) subsections describing the experiments, and a 3.1.3 subsection for analysing the results. Although, for consistency with the experiments with older children, the numbering of the sections describing the experiments and the ones analysing the results should follow the same schema, which is not the case even if you follow the corrections just described. In fact, the Results and the following sections have to be shifted by one, as there already exists a Section 3 (Ethical code).
All in all, I think it is an interesting contribution on the fundamental topic of STEAM education and the use of robots in particular.
Minor issues and typos:
-The Abstract says that experiments were performed with students aged 5 – 17 (line 39) whereas the ages in the Introduction are stated as ranging from 3 to 17 (line 61).
- “Testing of child robot interaction” (line 120) → Testing of child-robot interaction
- “Nevertheless, it has...” What follows does not contradict what has been previously stated: having a much wider range of movements is rather a consequence of the “added ability to move in two dimensions”. Therefore, instead of “Nevertheless”, I would use “Thus” (line 184).
- Title of Section 2.3: The word “Software” appears at the beginning and at the end of the title, I think the first one can be eliminated.
- Why is the complete name Lab Qui-Bot H2O used in line 204? It is a little bit confusing that the robots are sometimes named just Lab Qui-Bot, and in other occasions with the H2O appendix. If the “official” name includes H2O, but to avoid making the text too cumbersome you omit it, you should state this explicitly at he beginning and maintain the shortened version along the text. Otherwise one could think you refer to a specific component of the system.
- “2.5.3. D Qui-Bot Robot” → “2.5 3D Qui-Bot Robot” (line 312)
- “2x2x8x8 RGB LEDs” ? (line 332)
- Lines357, 358: The R and L batteries are the ones feeding the wheel actuators?
- Line 373 Missing space before the new sentence beginning with “Science…”. Also in line 391, between (I) and “fighting”.
- Lines 373-378: Make well-structured sentences, separate them with points, not semicolons.
- “ Quí-Bot” → Qui-Bot (lines 388 and 426)
- Lines 441-443 delete “should be presented”, as you already have a verb for this sentence (“teachers should explain and teach the reasons why…”).
- Line 490: There is already a Section 3, Ethical code (line 386).
- Line 498: “For girls and boys from 7 years, we provide” → from 7 to 17?
- “activities (One activity” → activities (one activity (line 499)
- “be revised as they are as they are commanded” → be revised as they are commanded (line 569)
- Line 597: What do you mean by “rapid use”?
- Line 605: What does k12 mean?
- First paragraph of Section 3.2 (lines 606-613): if there are just two activities, state it clearly. For example, “the following two activities: The first one was an activity for 30 girls… in [44]. As for the second activity, it consisted in a Codeathlon competition which was held for 50 boys and girls...”
- “named Qui-Bot H 2 O” (line 615): the confusion with the names continues. I mean, at this point the reader already knows the name of the robot, right? Does it refer to Lab Qui-bot, or to any of the other ones?
- Line 647: “with a 95% of confidence”→ You have just called this CI in line 645 (compare to 649, 650…).
- Lines 787-788: what kind of Theses? Graduate, Master, PhD?
- Line 827: “falsify” → refute, contradict, disprove…
- “and can have” → and have (line 839)
Author Response
All the issues have been addressed. Please see the attachment

Reviewer 3 Report
An interesting topic, important for improving the education of K12 students in the field of STEM. A well thought out and organized study. The methodological approach is adequate. Research (approach, methodology, organization, results, discussion) correctly presented. Great job! I encourage the authors to continue with further research, introducing new hypotheses and objectives. Also, he encourages the authors to examine the possibility of expanding the CDIO approach of engineering education to the K12 level, and in the theoretical domain, it would be worth trying to somehow connect the research method used, and the overall research context, with Kolb's experiential learning theory and the new taxonomy of educational objectives. In this way, your work, in addition to being professional, would certainly have a scientific dimension and possibly bring some concrete scientific contributions.
Author Response
We have addressed all comments. Please see the attachment.

Round 2
Reviewer 1 Report
Though the authors have pointed out the contributions in comparison with their earlier work which are not significant. Hence, the present manuscript has to be improved for further consideration.
